# Evaluation of Processing Conditions in the Performance of Purging Compounds for Polypropylene Injection Molding

Miguel Carrasco [1], Jorge Guerrero [2], Miriam Lazo [1,3], Estephany Adrián [1,3], Jorge Alberto Medina-Perilla [4] and Andrés Rigail-Cedeño [1,3,*]

1 Facultad de Ingeniería en Mecánica y Ciencias de la Producción (FIMCP), ESPOL Polytechnic University, Guayaquil P.O. Box 09-01-5863, Ecuador
2 Facultad de Ciencias Naturales y Matemáticas (FCNM), ESPOL Polytechnic University, Guayaquil P.O. Box 09-01-5863, Ecuador
3 Laboratorio de Procesamiento de Plásticos, ESPOL Polytechnic University, Guayaquil P.O. Box 09-01-5863, Ecuador
4 Materials and Manufacture CIPP-CIPEM Research Group, Mechanical Engineering Department, Universidad de los Andes, Bogotá 111711, Colombia
* Correspondence: arigail@espol.edu.ec

**Abstract:** Purging is a fundamental process in the injection molding sector, aiding in color transition, material shifts, and the removal of contaminants. The purging compounds can be classified according to physical or chemical mechanisms and are affected by processing parameters, such as temperature, pressure, or soaking period. Despite some studies on the effect of processing parameters in purging action, an analysis of the rheological behavior and physico-chemical changes is still required for a deeper understanding of this type of system. This study explored shear viscosity, activation energy behavior in the torque rheometer, injection molding process, and energy consumption for two polyolefin-based purging compounds: one on polypropylene (PP) and another on polyethylene (PE). The results showed that the PP-based compound is a highly viscous material with low thermal sensibility and low energy consumption. The PE-based chemical compound, which includes an expanding and scrubbing agent, presented higher thermal sensitivity. Lower purging times and specific energy consumption were observed for the mechanical purge regardless of the processing temperature in the injection molding machine. However, torque and specific total mechanical energy differed due to viscosity and possible filler particle agglomeration. These findings demonstrated the influence of processing temperature on rheology and performance. Nonetheless, further studies regarding pressure, soaking time, and rheological modeling are recommended.

**Keywords:** injection molding; purging compounds; rheology; activation energy

## 1. Introduction

Injection molding has become one of the dominant application segments for the polyolefin market, currently processing about 33% of polymers for end-user applications [1]. However, intermediate processes, such as preventive maintenance, mold shifts, and purging for color or material, drag down efficiency. The transitions from dark/intense to clear/transparent tones are the most challenging color changes during plastic injection molding. As a response, some industrial practices have been implemented, such as acetylene torching or manual screw cleaning with brass gauze and stearic acid. However, these methods are time-consuming and might affect the metal properties [2]. Therefore, purging compounds have been developed to clean contaminants formed during plasticization [3] and to prevent the accumulation of colored material, degradation, blackspots, and oxidized gels [4,5].

Purging compounds are used to remove the remaining material from the barrel through alternative mechanisms: mechanical purges, chemical purges, resin-based surfactants [5], and regrinds [6]. Mechanical purging compounds may use abrasive fillers or

high-viscosity polymers to scrub or push contaminants out of the barrel [7,8]. Regrind or recycle usage is a traditional approach, but it is less effective, becoming expensive in the long run. Chemical purges react with the residue to cause depolymerization, lowering its viscosity and facilitating evacuation [4]. Surfactant-filled resins are thermally stable additives that loosen build-up by forming anionic or non-ionic bonds [5]. Therefore, depending on production rates, color shifts, and the polymer base, an appropriate purge compound may be relevant for productivity.

The favoring parameters for mechanical commercial purging compounds (CPC) are high screw speeds and back pressures that enhance the scouring of internal metal surfaces. Chemical compounds have been suggested to act better upon increased soaking time and temperature [9]. Nonetheless, soaking time increases machine downtime [10], and excessive temperatures may cause degradation. In both cases, viscosity and flow variations are required. Therefore, it is necessary to determine the effects of key parameters in the purging process using different compounds.

Available polymer-based purging compounds use diverse materials from polyolefins [11] to fluoropolymers [12] and polymer blends such as linear low-density polyethylene and ethylene methyl methacrylate [13]. However, the selection of the CPC involves different factors [14]:

- Type of polymer to be purged;
- Process temperature range;
- Polarity;
- Type of plastic conversion process.

Consequently, CPC manufacturers offer a wide range of purging products and even offer assistance and tailor the purge for their clients [15].

Although different CPCs are available in the market, injection molders tend not to follow the methodologies suggested by manufacturers creating their own solutions, such as diluting CPC with virgin resin. As a result, cost, time, and CPC effectiveness are variable [16,17]. Different studies have focused on showing the effect of CPC on the residence time distribution [3,10], comparing time and temperature dependence on the purging process with and without CPC [18], and assessing the processing parameters using hybrid CPC to achieve enhanced purging efficiency [10]. However, a thorough characterization analysis of CPC active agents, such as abrasives, reactants, or surfactants, to explain the influence of temperature on mechanical and chemical CPC performance has not been deeply explored.

Specific energy consumption (SEC) has been a relevant consideration in plastic processing due to its inherent energy demand in manufacturing. A study comparing commercial purging compounds against virgin resins demonstrated a cost saving of 73–83% [16], which considered material usage, purging time, and energy consumption. Although several studies have been performed on injection molding to optimize the overall process [19], SEC has hardly been considered for these purging studies. Additionally, previous research has emphasized torque rheometry's importance in understanding unknown materials' rheology and energy consumption [20]. Hence, an exhaustive SEC analysis must be performed to determine the sustainability performance of purging materials in an extensive scale manufacturing process.

In 2021, polypropylene and polyethylene (low and high-density) reached the top three in plastic converters demand [21] and the most imported plastics in countries such as Ecuador [22], which only perform plastic transformation processes. Thus, this study aimed to determine the rheological behavior of two different polyolefin-based CPC and the effect of the processing temperature on the purging performance. The incidence of the CPC composition in the process was also considered. For this, the viscosity of commercial purging compounds, the characterization of active agents contained in the CPC, and the effect of temperature were analyzed. Finally, the SEC was correlated to processing conditions to establish a sustainable performance for purging in injection molding.

## 2. Research Methods and Equipment

### 2.1. Materials

ASACLEAN UP Grade, a polypropylene-based mechanical purge (PP-MP), was supplied by Asaclean® Purging Compounds (Parsippany-Troy Hills, NJ, USA). Kalay Ultra Plast PO-E polyethylene-based chemical purging compound (PE-CP), constituting a polymer resin and an external ceramic-like filler. Braskem Polypropylene (PP) Homopolymer H 202HC and purging compounds were donated by PICA Plásticos Industriales C.A. (Guayaquil, Ecuador). Table 1 displays technical information regarding the materials employed.

**Table 1.** Purging compounds and PP information provided by manufacturers.

| Material | Density [g·cm$^{-3}$] | Melt Flow Rate (MFR) [g/10 min] | Recommended Process Temperatures [°C] |
|---|---|---|---|
| PP-MP | 1.09 (23 °C) | N/A | 170–300 |
| PE-CP | 0.70 (25 °C) | N/A | 140–300 |
| PP | 0.905 (23 °C) | 23 [a] | N/A |

[a] MFR at 230 °C/2.16 kg.

### 2.2. Characterization of Purging Compounds

#### 2.2.1. Fourier Transform Infrared Spectroscopy (FTIR)

Fourier transform infrared spectroscopy was conducted in a PerkinElmer Spectrum 100 spectrometer in the Mid-IR absorption spectra ranging from 4500 to 450 cm$^{-1}$ to determine information about the functional groups present in the PE-CP external filler.

#### 2.2.2. X-ray Diffraction (XRD)

X-ray diffraction was evaluated using a PANalytical XPert-Pro diffractometer, Co K$\alpha$ ($\lambda$ = 1.78901 Å), radiation operated at 45 kV and 30 mA with a 1/8″ divergent slit, and 1/16″ anti-scatter slit. The scanning ranged from 2.5° to 90° (2θ), 0.05° step size, 20 s per step. Compound identification was performed using a search/match algorithm of X'Pert HighScore Plus Version 2.2.3 PANalytical B.V. software (Almelo, The Netherlands).

#### 2.2.3. Thermogravimetric Analysis (TGA)

Thermal stability analysis was performed in a TA Instruments Thermogravimetric Analyzer Q600 STD (TGA/DSC) at a heating rate of 10 °C·min$^{-1}$ in a nitrogen atmosphere (100 mL·min$^{-1}$) from room temperature to 800 °C. The residues' microscopic structure was observed using a Wild Heerbrugg M400 Photomakroskop and a Sony Exmor EC-MOS03100KPA industrial digital camera.

#### 2.2.4. Differential Scanning Calorimetry (DSC)

Thermal properties, such as specific heat capacity, melting, and crystallization, were examined in TA Instruments Q200 DSC analyzer.

Double-run experiments were conducted at a temperature range from 23 °C to 250 °C under a nitrogen atmosphere (50 mL·min$^{-1}$) with a heating rate of 10 °C·min$^{-1}$. Crystallinity fraction ($X_c$) was measured using melting enthalpy ($\Delta H_m$), as seen in Equation (1) [23]. Theoretical fusion enthalpies ($H_m^o$) for completely crystalline PP and HDPE are 207 J·g$^{-1}$ [24] and 293 J·g$^{-1}$ [20], respectively.

$$X_c = \frac{H_m}{H_m^o} \tag{1}$$

Before specific heat capacity testing, temperature calibration was performed with a sapphire disc. Then, samples were sealed in a hermetic aluminum pan and equilibrated at 0 °C for 5 min, followed by heating to 270 °C, with a ramp of 3 °C·min$^{-1}$, and modulation of 1 °C every 100 s.

### 2.3. Rheological Analysis

The rheological behavior of purging compounds was studied in a Brabender Plastograph® EC Plus torque rheometer. A total of 35 g of purging compound were fed into the mixing chamber and processed for 10 min at different temperatures (220 °C, 240 °C, and 260 °C) and screw speeds (10, 30, 50, 70, and 90 rpm).

The torque rheometer provided information about the additives' flow behavior and apparent viscosity. Additionally, it gives an overall idea of the shear rate, temperature, and energy required from the additives in optimizing polymer processing. Therefore, torque and rotor speed has been transformed into rheological data using the Newtonian approach presented by Bousmina [25]. The internal radius has been considered independent of the fluid nature, and the shear rate $(\dot{\gamma})$ is calculated through Equation (2):

$$\dot{\gamma} = \frac{2\pi N}{\ln \beta}, \tag{2}$$

where $N$ is the angular speed of the rotor (s$^{-1}$); $\beta$ is the ratio between the external and internal radii (*Re/Ri*). The viscosity $\eta$ is given by Equation (3):

$$\eta = \frac{\Gamma(\beta^2 - 1)}{N8\pi^2 LRe^2(1 + g^2)}, \tag{3}$$

where $\Gamma$ is the torque (N·m); $L$ is the length of the cylinder; $Re$ is the external radius; and $g$ is the gear ratio taken as 2/3. The dimensions determined by the calibration of the equipment are detailed in Table 2.

**Table 2.** Torque rheometer dimensions.

| Section | Size [cm] |
|---|---|
| Internal Radius | 1.65 |
| External Radius | 1.85 |
| $\beta$ | 1.12 |
| Cylinder Length | 4.60 |

On the other hand, to obtain the flow activation energy ($\Delta E$), the Arrhenius relationship of $\eta$ and the temperature [26] should be considered:

$$\eta = Ae^{\Delta E/RT}, \tag{4}$$

herein, $A$ is the Arrhenius constant; $R$, the universal gas constant (8.314 J·mol$^{-1}$·K$^{-1}$); and $T$ is the absolute temperature. Thus, the following association is obtained by logarithmically linearizing Equation (4):

$$\ln \eta = \frac{\Delta E}{R} \left(\frac{1}{T}\right). \tag{5}$$

The flow activation energy is the amount of energy polymer chains require to begin flowing when the temperature increases [27]. In other words, $E$ (expressed in J·mol$^{-1}$) represents the degree of thermal sensibility that a material shows during heating at a determined temperature range. However, $E$ might decrease at higher temperature intervals because molecules move freely, so less energy is required. Consequently, the effect of temperature in the purging process improves and optimizes the temperature conditions for each purging compound.

The polymers' processability is connected to the material's production energy requirements and flow properties toward its transformation into a final product [28]. It is a complex index often related to viscous, thermophysical, and mechanical properties, among other parameters, which depend on the processing method [29]. Based on the mentioned concepts, processability has been represented by maximum torque and the specific total

mechanical energy (*STME*) required for processing at 90 rpm. The *STME* is given by Equation (6).

$$STME = \frac{N \int Mdt}{m}, \tag{6}$$

where $\int Mdt$ refers to the integral under the curve for the torque-time plot in N·m·s, $N$ is the screw speed in $s^{-1}$, and $m$ is the mass of the sample in kg.

### 2.4. Purging Methodology

Table 3 shows the time injection parameters, such as the injection time and cycle, for PE-CP and PP-MP compounds. The injection molding process was performed using a Lien Yu D75 machine (Figure 1a) with four heating zones, a screw diameter of 75 mm, a 20 L/D screw ratio, a maximum shot weight of 115 g, and a maximum injection pressure of 1777 bar. The back pressure was fixed at 14.71 bar, and the screw speed was 60 rpm. Although the mechanisms of each purging compound differ, a general procedure follows the same steps.

**Table 3.** Average injection and cycle times [s] for each purging compound at different temperatures.

| Purging Compound | Injection Process | Temperature Profile | | |
| --- | --- | --- | --- | --- |
| | | **220 °C** | **240 °C** | **260 °C** |
| PE-CP | Injection time | 3.68 | 3.97 | 3.73 |
| | Cycle | 33.15 | 29.65 | 28.10 |
| PP-MP | Injection time | 3.53 | 3.57 | 3.62 |
| | Cycle | 24.86 | 24.93 | 25.08 |

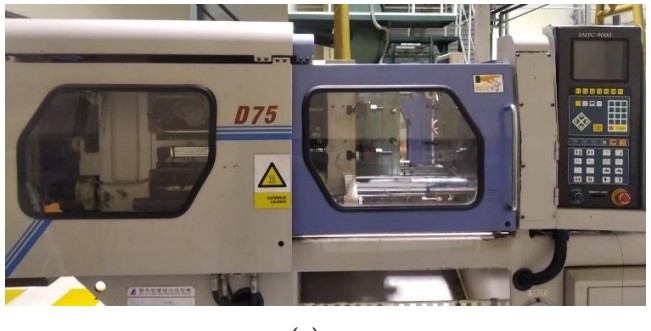

(**a**)

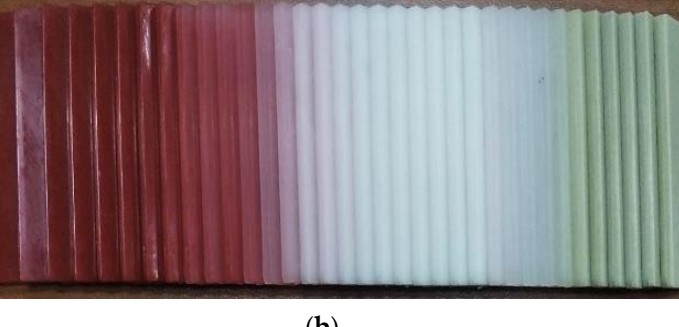

(**b**)

**Figure 1.** (**a**) Injection molding unit and (**b**) Injection molded pieces: sequential color shifts from red to beige for PP-MP at 220 °C.

Red pigmented PP (200 g) was introduced into the injection molding process, followed by virgin PP (200 g) to simulate the transition from a darker color. Afterward, 300 g of one of the purging compounds was added to the feeding zone and injected until the barrel was emptied. Finally, non-pigmented PP (200 g) and beige PP (200 g) were added sequentially. An academic test specimen mold was used solely to observe the color transition on injected pieces.

Table 4 presents the three temperature profiles that were used to perform the purging process with each compound. Temperatures were selected considering 300 °C as the maximum processing temperature for each CPC [11,30] and the PP injection molding temperature range [31]. Purging periods were taken from the CPC feeding until fully beige-injected pieces were obtained. An analysis of variance (ANOVA) was applied to five experiments performed at three different temperatures. ANOVA allowed us to measure the effectiveness of the process temperature on the purging time. This variable was chosen because optimizing the downtime during purging is vital for the plastics injection molding

industry. The sum of squares (S.S.), mean square (M.S.), and F-values were calculated with a confidence level of 95%.

**Table 4.** Temperature profiles for injection molding purging.

| Zone | Purging Temperature Profiles [°C] | | |
|:---:|:---:|:---:|:---:|
| | 1 | 2 | 3 |
| 1 | 220 | 240 | 260 |
| 2 | 220 | 240 | 260 |
| 3 | 200 | 200 | 200 |
| 4 | 190 | 190 | 190 |

*2.5. Specific Energy Consumption*

The mathematical model, proposed by Elduque et al. [32], was considered to estimate the energy consumption during the purging process. Under this scheme, the parameters used for its construction were the machine's utilization and efficiency, throughput, and polymer features as follows:

$$SEC \left( \frac{kWh}{kg} \right) = (7.5 - (5 \times \left( \frac{E}{100} \right) \times \left( \frac{\left( \frac{w \times \frac{3.6}{t_c}}{0,0052 \times F_c} \right)^{0.15}}{C_{sp} \times \left( \frac{T_i - T_a}{305.255} \right)^{0.1}} \right) \times \left( w \times \frac{100}{\rho \times V_{max}} \right)^{-0.5}, \quad (7)$$

$E$ is the machine's efficiency; $w$ is the mass (115 g) of the injected piece; $t_c$ is the cycle time (s) at different temperatures; $F_c$ is the clamping force of 735.5 kN, and $C_p$ is the specific heat of the material. $T_i$ and $T_a$ are the injections and room temperatures, respectively. In this case, $T_a$ was fixed to 25 °C, and $T_i$ was assumed to be 220, 240, and 260 °C. Finally, $\rho$ is the density of the injected material in g·cm$^{-3}$, and $V_{max}$ is the maximum injection volume in cm$^3$. A value of 128 cm$^3$ was assumed according to the specified machine's swept volume. The density was estimated given the polymer base as 0.905 g·cm$^{-3}$ and 0.95 g·cm$^{-3}$ [33] for PP-MP and PE-CP (without filler), respectively.

**3. Results and Discussion**

*3.1. Characterization of Purging Materials*

3.1.1. FTIR and XRD Analysis of PE-CP Filler

PE-CP comprises a polyethylene (PE) matrix and a filler that acts as an expanding and scrubbing agent once it reaches specific processing temperature windows [34]. This filler may contain a binder, a surfactant, a blowing agent, and an inorganic scrubber [34]. Figure 2 presents PE-CP's filler spectroscopy. The results show different regions associated with amine, hydroxy, and aliphatic groups, as seen in Table 5. A broad-strong peak was attributed to -OH stretching at about 3434 cm$^{-1}$. In addition, two sharp medium peaks of primary amine stretching were seen at 3697 cm$^{-1}$ and 3621 cm$^{-1}$ and might correspond to the ester-amide binder or blowing agents [34]. As blowing agents undergo the gas release of $CO_2$ or $N_2$ to induce resin foaming, they may be composed of organic acids, nitrate compounds, nitrogenated organic compounds, and others.

Surfactants are natural/mineral oils, and binders are waxes [35], usually employed in chemical purge materials. In this context, the surfactant has essential functions in the filler, such as the external lubrication of the CPC [36], the dispersion of the inorganic particles [37], and contaminants softening [38]. Effective processing surfactants are usually long-chain fatty acids, metal soaps, or molecules with a strong affinity for superficial metal cations. For instance, stearic acid is a typical coating for calcium carbonate particles [39] and has been recommended to loosen build-ups in plasticizing units.

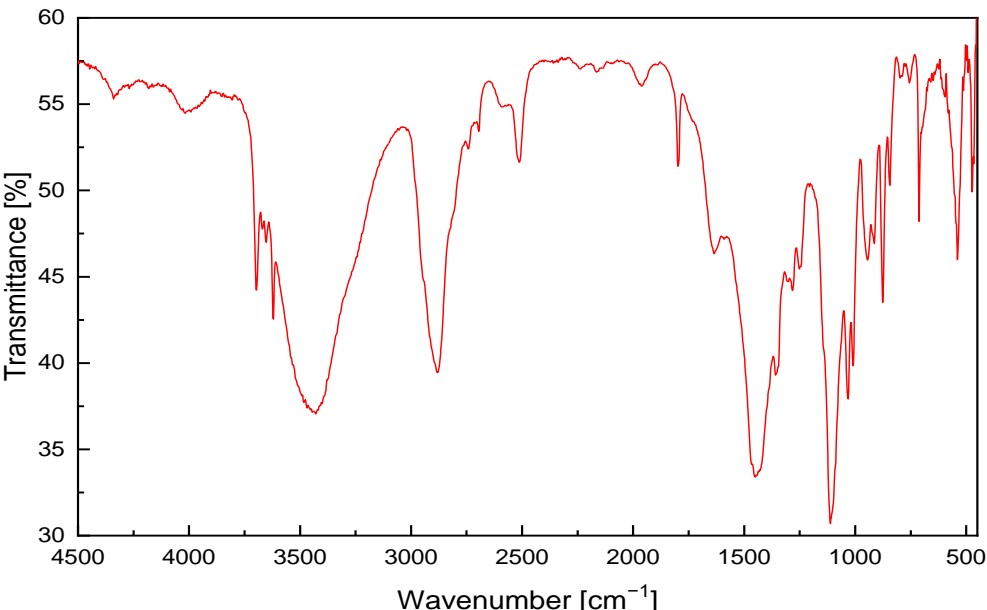

**Figure 2.** FTIR spectrum for PE-CP filler.

**Table 5.** FTIR peaks for PE-CP filler.

| IR Peak [cm$^{-1}$] | Functional Group |
|---|---|
| 3696/3621 | –NH$_2$ stretching |
| 3434 | –OH stretching |
| 2919 | –CH$_3$ asymmetric vibration |
| 1727 | C=O aliphatic carbonyl stretch |
| 1633 | –NH bending |
| 1413 | –OH bending |

On the other hand, the binder is a wax typically composed of hydrocarbon chains or ester-amide [35]. The aliphatic vibration region at about 2900 cm$^{-1}$ may be associated with the methylene and methyl groups in the binder and surfactant components of chemical purges. In the 2000–1500 cm$^{-1}$ region, a carbonyl stretch is shown at 1727 cm$^{-1}$ and is possibly associated with acids or ester-amides in the filler. Additionally, the peaks at 1413 cm$^{-1}$ and 1100 cm$^{-1}$ correspond to –OH bending and –COC stretching, respectively, of carboxylic acid or ether groups in the compound.

Figure 3 shows the XRD pattern for PE-CP filler inorganic groups. Inorganic abrasive elements for CPC may include clays, calcium carbonate, silica, and talc [13]. The XRD search and match algorithm indicates a higher probability of calcium carbonate in the form of calcite than kaolinite or silica. Calcium carbonate is often used as an abrasive or scrubbing component for mechanical and chemical purges [40].

3.1.2. Thermogravimetric Analysis

Figure 4 illustrates TG and DTG curves for purging compounds and the PE-CP filler. PP purge experienced a two-step degradation. Between 350 and 500 °C, the first stage represented a weight loss of 76.2% with a high peak at 459.86 °C, closely related to the chain scission mechanism during polyolefin's thermal decomposition [41], reflected on the PP TG curve. From 500 to 800, the degradation phase corresponded to a 7.2% material loss, leaving a 16.6% residue. In addition to the PP base, the mechanical CPC may include processing aids or abrasive fillers [40].

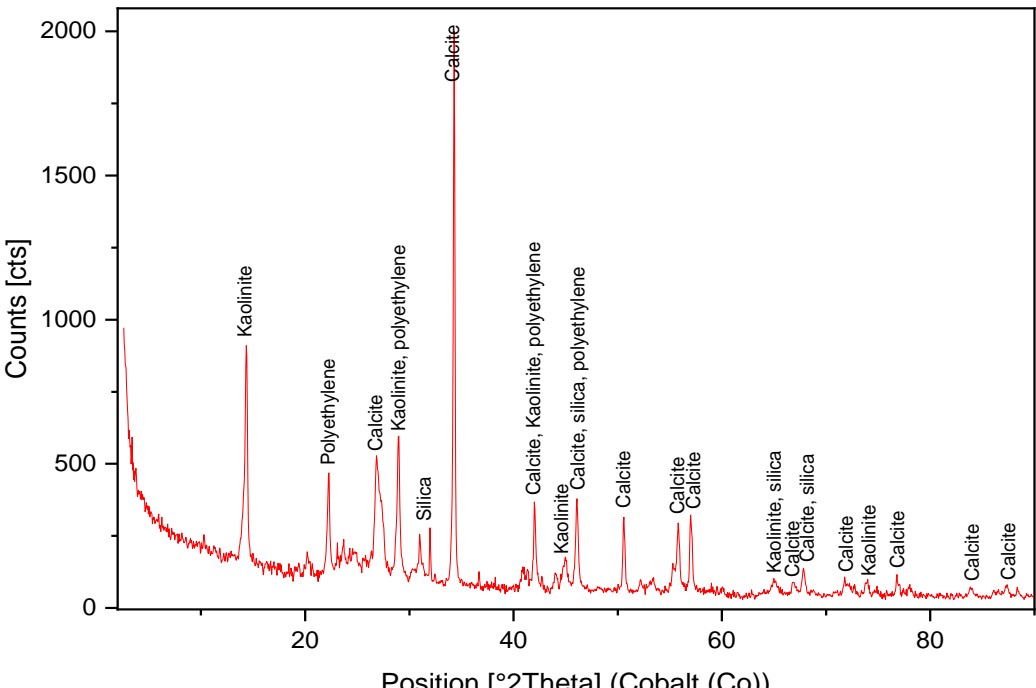

**Figure 3.** XRD profile for PE-CP filler.

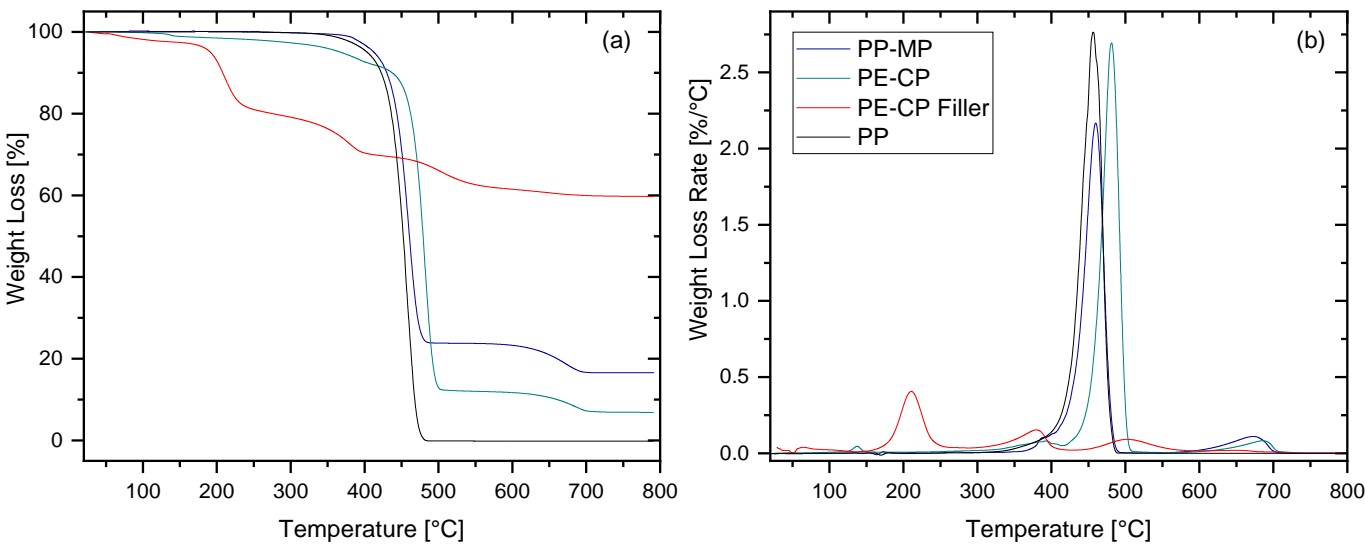

**Figure 4.** TG (**a**) and DTG (**b**) temperature profiles for purging compounds, PE-CP filler, and PP.

On the other hand, the neat polyethylene-based purging compound displayed four degradation stages. The highest peak temperature at 481.54 °C comprised the polymer matrix decomposition between 400 and 600 °C [20]. Weight loss changes of 1.6, 7.2, and 4.8%, and a residue of 6.9% indicated the presence of different agents within the compound. For example, hybrid purges [40] may employ weak metal adsorbates to remove contaminants chemically.

Figure 5 displays the microscopic structures of PP-MP and PE-CP after thermal degradation at 800 °C. The PP purging compound's residue appeared to be ceramic-like white dust. On the contrary, the PE chemical purge turned out to be a black-porous residue. Both presented glassy structures that may have come from the vitrification process of ceramic components at high temperatures [42] or crystal structures from other elements.

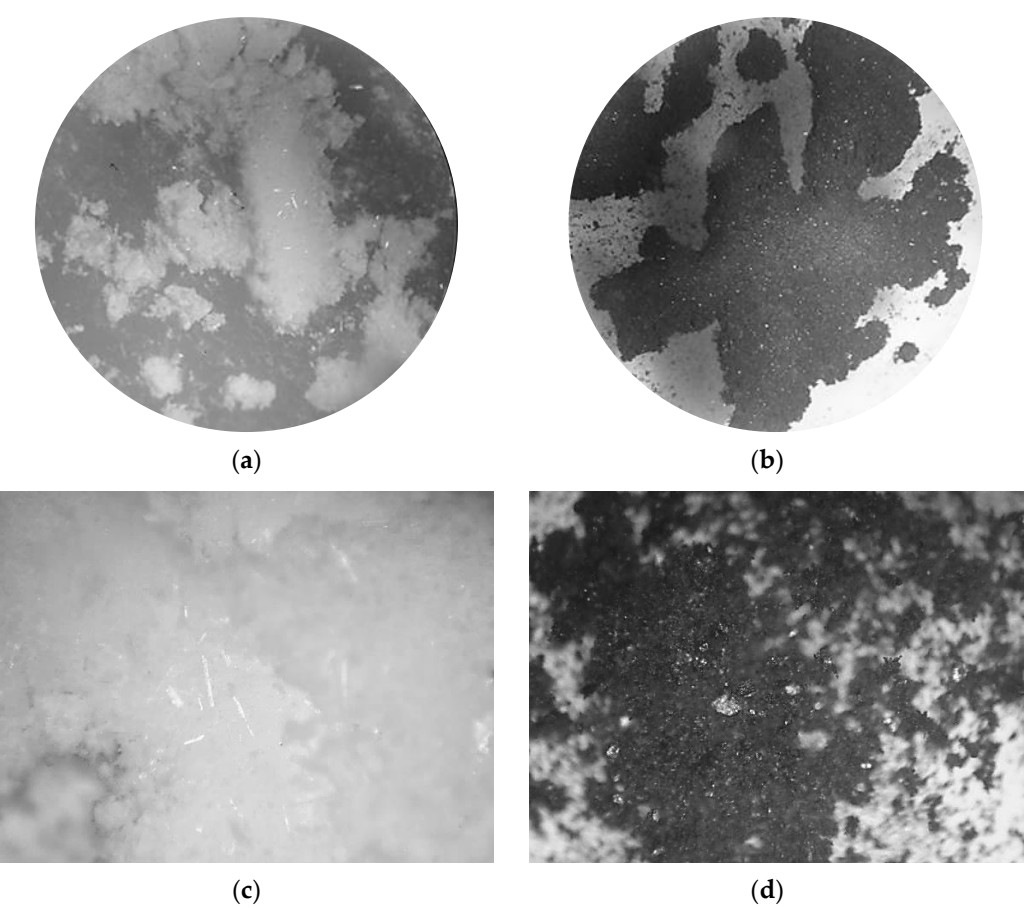

**Figure 5.** Microscopic observations for PP-MP (left) and PE-CP (right) residue after thermal degradation at (**a**,**b**) 32× and (**c**,**d**) 100× magnification.

The PE-CP filler's thermal profile shows four significant mass losses during heating. The first mass loss step of 2.395% at 64.05 °C is related to absorbed moisture or volatiles. The second thermal degradation, between 150 and 280 °C, represents a weight loss of 17.69% at 210.59 °C. The thermal decomposition of blowing agents, for instance, citric acid and sodium bicarbonate, usually revolves around this temperature window [43]. The following mass reduction of 10.27% could be related to the decomposition of the wax or surfactant at about 380 °C. The 59.73% residual components are considered to be inorganic scrubbing agents, unaffected above 800 °C.

### 3.1.3. Differential Scanning Calorimetry

Table 6 shows the DSC results for the purging additives and virgin PP. PE-CP underwent melting and crystallization, as its polymer matrix, at 136.47 °C and 112.79 °C, respectively. On the other hand, PP-MP shared a similar thermal behavior with virgin PP. The PP mechanical purge presented a fusion peak at 165.43 °C and crystallized at 125.33 °C. Nevertheless, an endothermal process was observed at 131.02 °C. The signal might be associated with polyethylene traces in the purging compounds formula [44].

**Table 6.** DSC melting and crystallization parameters for purging compounds and virgin PP.

| Material | $T_m$ [°C] | $T_c$ [°C] | $\Delta H_m$ [J·g$^{-1}$] | $X_c$ [%] |
|---|---|---|---|---|
| PP-MP | 165.43 (131.02) | 125.33 | 45.26 (5.28) | 21.86 |
| PE-CP | 136.47 | 112.79 | 148.30 | 50.61 |
| Virgin PP | 164.68 | 134.17 | 114.10 | 55.12 |

Values in parenthesis correspond to melting temperatures and enthalpy of a material different from polypropylene in the PP mechanical purge.

Virgin PP melting and crystallization temperatures were 164.68 °C and 134.17 °C, respectively. The material was 55.12% crystalline. Although the crystallinity of PP-MP is lower than PP homopolymer, the effect of crystallinity on the polymer's rheological interaction does not seem relevant for purging.

Table 7 displays the specific heat capacity for both purging compounds at three different temperatures. The heat capacity of polymers increases with temperature and is related to the vibrations of the backbone and substituent groups [45]. Comparing Cp results, the mechanical purge requires less energy to raise its temperature than PE-CP: a phenomenon related to each base polyolefin thermal behavior and crystallinity degree.

**Table 7.** Specific heat capacity [kJ·kg$^{-1}$·K$^{-1}$] for the purging compounds at different temperatures.

| Temperature [°C] | PE-CP | PP-MP |
|---|---|---|
| 220 | 3.682 | 1.824 |
| 240 | 4.119 | 2.448 |
| 260 | 4.717 | 2.511 |

*3.2. Rheological Analysis*

Tables 8 and 9 display the STME and stabilized torque values for PP-MP and PE-CP at different shear rates and temperatures. In both purges, the energy decreased with higher temperatures but increased with the rotor speed. The STME equation depends linearly on the rotor speed and the area under the torque vs. time curve. The lowest energy is observed at high temperatures and low rotor speeds since the screw exerts less effort while mixing due to the compounds' lower viscosity. However, PP-MP requires more energy during the first minute as a narrow peak area appears in the Torque–rheometer curve, as observed in Figure 6b.

**Table 8.** Specific energy and maximum torque at different mixing conditions of PP-MP.

| Rotor Speed [rpm] | 220 °C | | 240 °C | | 260 °C | |
|---|---|---|---|---|---|---|
| | STME [kWh·kg$^{-1}$] | Stabilized Torque [N·m] | STME [kWh·kg$^{-1}$] | Stabilized Torque [N·m] | STME [kWh·kg$^{-1}$] | Stabilized Torque [N·m] |
| 10 | 0.03 | 4.81 | 0.03 | 3.57 | 0.02 | 3.18 |
| 30 | 0.11 | 4.67 | 0.09 | 4.26 | 0.08 | 3.89 |
| 50 | 0.19 | 4.75 | 0.16 | 4.41 | 0.15 | 3.94 |
| 70 | 0.25 | 4.50 | 0.22 | 4.26 | 0.20 | 3.78 |
| 90 | 0.31 | 4.20 | 0.27 | 3.16 | 0.26 | 2.94 |

**Table 9.** Specific energy and maximum torque at different mixing conditions of PE-CP.

| Rotor Speed [rpm] | 220 °C | | 240 °C | | 260 °C | |
|---|---|---|---|---|---|---|
| | STME [kWh·kg$^{-1}$] | Stabilized Torque [N·m] | STME [kWh·kg$^{-1}$] | Stabilized Torque [N·m] | STME [kWh·kg$^{-1}$] | Stabilized Torque [N·m] |
| 10 | 0.01 | 1.86 | 0.01 | 1.65 | 0.01 | 1.27 |
| 30 | 0.06 | 3.84 | 0.05 | 3.50 | 0.04 | 2.82 |
| 50 | 0.12 | 4.40 | 0.11 | 4.05 | 0.07 | 2.75 |
| 70 | 0.20 | 5.23 | 0.17 | 5.22 | 0.12 | 3.51 |
| 90 | 0.29 | 5.52 | 0.22 | 4.31 | 0.19 | 4.37 |

Stabilized torque refers to the torque value at a steady state at the end of the mixing time. A slight decrease was observed for the torque trend in PP-MP with higher rotor speed under a fixed temperature, which could be related to the viscosity reduction due to shear heating. Shear heating or viscous dissipation is formed by the friction between polymer particles and the material within the mixing chamber [46], resulting in a temperature increase. On the other hand, PE-CP's torque behavior might be influenced by the formation

of clusters of rigid inorganic particles in the mixture because of the dipole–dipole interaction and low polymer-particle compatibility [47]. Hence, as shear increases, networks of small particles fill larger interparticle gaps, and the torque increases [48]. However, the polyethylene-based compound behaves similar to a temperature-sensitive, shear-thinning fluid due to the superposition of the polymer melt matrix [49] and filler disintegration.

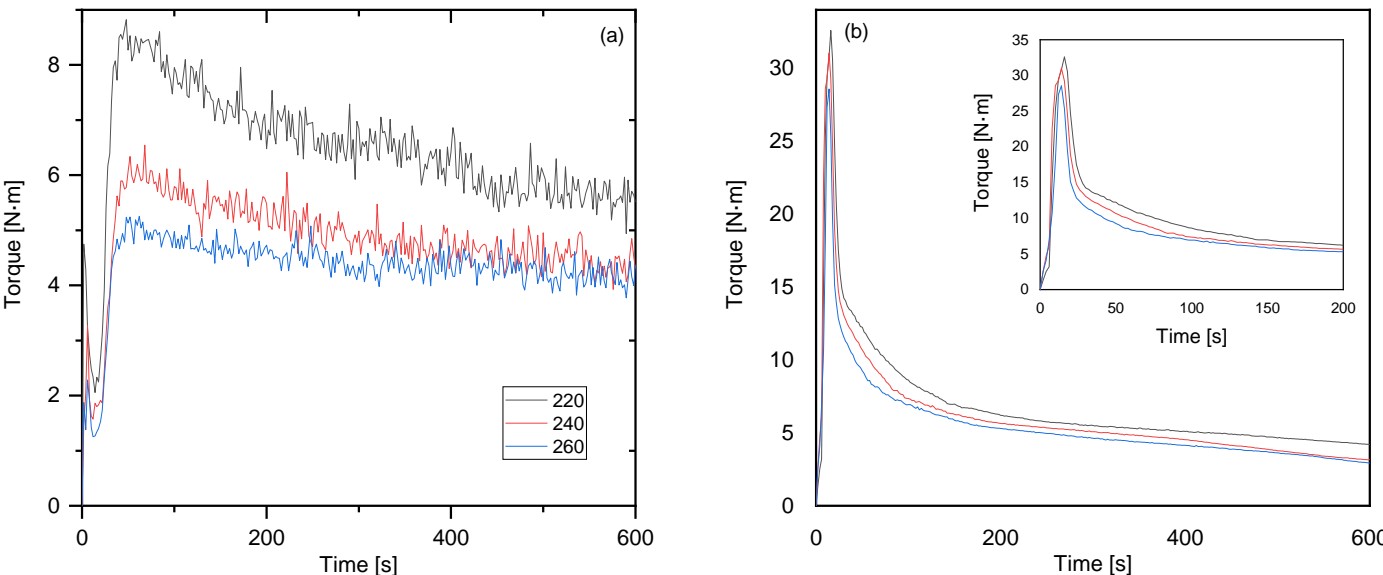

**Figure 6.** Torque–time curve for (**a**) PE-CP and PP-MP (**b**).

Figure 6 shows the torque–time curves for both purging compounds at 90 rpm. For PE-CP, there are two peaks within the first minutes: the first one might be associated with the filler expansion in the feeding zone, and the second one to a viscosity increase before the polymer's fusion. The maximum torque at 90 rpm was 8.82 N·m at 220 °C. Herein, the compound's fusion behavior is related to the thermal energy transfer from the rheometer chamber, so higher temperatures will reduce the sample's viscosity in less time [50]. PP-MP curve presented a single maximum torque within the first minutes of mixing and melting. Furthermore, PP-MP's maximum torque was 32.595 N·m at 220 °C.

The variations in torque for both purging compounds align with their different compositions and purging mechanism principles. The chemical purge relies upon scrubbing and expanding functionalities of its filler rather than a viscosity difference. Torque stability is reached faster using the mechanical purge instead of the chemical one. This may occur due to substances such as calcium carbonate particles in PE-CP's mixture, which raises viscosity because of its agglomeration [50] or wall-slip due to external lubrication by wax or surfactant.

The torque rheometer data were converted into rheological information by applying the Bousmina model [25]. Thus, the viscosity values were used to calculate the flow activation energy and create viscosity versus shear rate curves. Table 10 displays the activation energy at different experiment screw speeds. According to data, increased shear rates slightly reduced the energy activation for both purging compounds due to the free volume increment or alignment in polymer chains [51]. Additionally, it has been reported that the flow activation energy is influenced by chain flexibility [52], molecular weight [53], intermolecular interactions [54], and free volume obstruction [55], especially in inorganic particle-filled polymer composites such as PE-CP. Therefore, PE-CP requires higher amounts of energy than PP-MP as it is more sensitive to temperature. Finally, contrary to previous observations, neat PP yields higher energies with a shear rate.

**Table 10.** Flow activation energy [kJ·mol$^{-1}$] for purging compounds at constant shear rates.

| Speed [rpm] | PP-MP | PE-CP |
|---|---|---|
| 10 | 18.35 | 33.13 |
| 30 | 12.17 | 19.23 |
| 50 | 9.16 | 25.22 |
| 70 | 6.56 | 26.94 |
| 90 | 10.35 | 19.83 |

Figure 7 presents the viscosity values for the purging compounds at three temperature profiles. Figure 7a shows that PP-MP's apparent viscosity decreases with heating at low shear rates. Nonetheless, the reduction in viscosity due to temperature rise is negligible at higher shear rates. On the contrary, the downward shift between PE-CP isotherms is evident throughout the shear rate range, as seen in Figure 7b, implying that both compounds have shear thinning behavior but different thermal susceptibility. Furthermore, base polypropylene has a lower viscosity than both purging compounds, according to Figure 7c. This viscosity difference suggests that PP-MP and PE-CP could perform using a viscosity gradient to a certain degree.

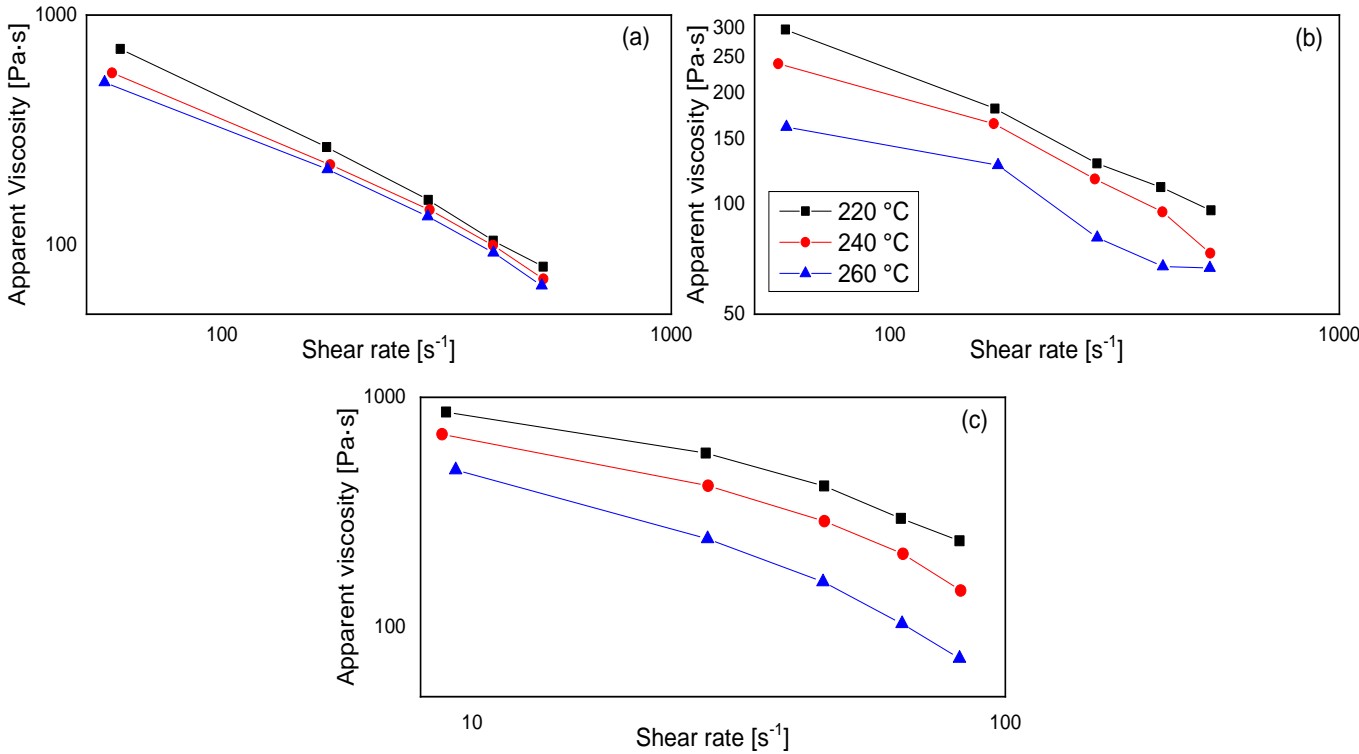

**Figure 7.** Apparent viscosity vs. shear rate (**a**) PP-MP; (**b**) PE-CP; (**c**) Virgin PP.

### 3.3. Statistical and Specific Energy Consumption Analysis in Injection Molding

Table 11 presents the mean purging times for each compound at different injection temperatures. For PE-CP, a temperature increase from 220 °C to 260 °C reduced the time needed for purging, which is essential for injection molders to comply with their schedule and reduce energy consumption. In contrast, the purging times of PP-MP grew subtly with temperature. However, the PP-MP purging periods were still lower than PE-CP at every studied temperature. In general, the temperature positively influences the purging performance of the PE-CP but is less relevant for PP-MP.

**Table 11.** Mean purging times [min] for each injection temperature.

| Purge | Injection Temperatures [°C] | | |
|---|---|---|---|
| | **220** | **240** | **260** |
| PE-CP | 18.48 | 19.20 | 14.74 |
| PP-MP | 13.26 | 13.30 | 13.38 |

ANOVA application corroborates the time correlation between purging compounds against temperature. ANOVA was used to examine the significance of temperature over purging time, fixing a 95% confidence level. The sum of squares, mean square, and Fisher's value were calculated for each purging compound, as shown in Table 12. The *p*-values under 0.05 indicate that the factor is significant, and the null hypothesis, all means are the same, is rejected. The *p*-value was ~0 and 0.392 for PE-CP and PP-MP, respectively. When comparing the contribution percentages of temperature in both purges, PE-CP purging performance is more sensitive to heating than PP-MP. A similar tendency was also observed in the heat capacity analysis.

**Table 12.** Statistical results for purging methodology.

| Purge | Control Factor | Degrees of Freedom | Contribution [%] | Sum of Squares | Mean Square | F | *p* |
|---|---|---|---|---|---|---|---|
| PE-CP | Temperature | 2 | 76.79 | 36.61 | 17.80 | 19.85 | 0.0002 |
| | Error | 12 | 23.21 | 10.76 | 0.90 | | |
| | Total | 14 | 100.00 | 46.37 | | | |
| PP-MP | Temperature | 2 | 14.44 | 0.037 | 0.018 | 1.01 | 0.3922 |
| | Error | 12 | 85.56 | 0.22 | 0.018 | | |
| | Total | 14 | 100.00 | 0.25 | | | |

Figure 8 shows the specific energy consumption for each purging additive. According to the SEC model, energy consumption depends on time, temperature, specific heat, and the material's density [32]. PE-CP purging time shortens with the temperature rise, as seen in Table 12. Nonetheless, its physical properties, such as density and specific heat, increase SEC. In contrast, the PP-based mechanical compound performs for shorter times and has a lower density that reduces the specific energy consumption in the injection molding equipment.

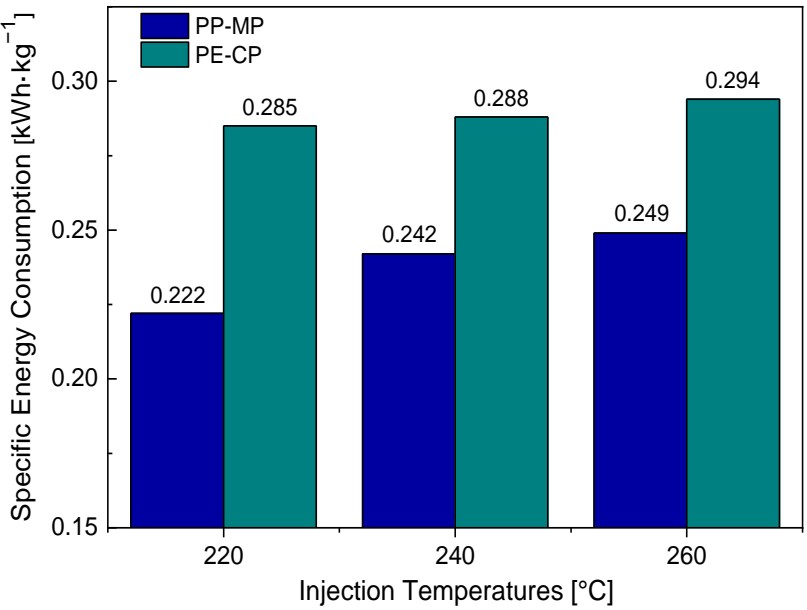

**Figure 8.** Specific energy consumption of purging compounds at different injection temperatures.

## 4. Conclusions

The evaluation of the effect of temperature on the rheology and performance of purging compounds for polyolefin injection molding was determined. Two purging compounds were analyzed: one PP-based and the other PE-based. The first PP-MP is an amorphous mechanical purge with low thermal sensibility, high viscosity, and apparent shear-thinning behavior. In contrast, the PE-CP is formed by a thermally sensitive semicrystalline polyolefin base and a filler that undergoes various degradation reactions, enabling it to expand and scrub the build-ups. The characterization studies suggested that the filler contains an expansion agent compacted with wax and surfactants.

Regarding processability, PP-MP presented higher torque and STME than PE-based compounds along the plasticization process in the torque rheometer. On the other hand, the purging performance has been represented by energy consumption and purging time in the injection molding machine. The first one is influenced by the physical properties of the purges, such as density and specific heat, but their differences become less evident with increasing temperature. The purging time is lower than PE-CP's and statistically unaffected by temperature, which is coherent with their purging mechanisms.

Finally, further studies are required regarding the effect of pressure and soaking time on the purging performance of mechanical and chemical compounds. It is also recommended to evaluate the behavior of the materials over a broader range of shear rates that lie on the injection molding region. Additionally, a modeling study should be performed to identify a suitable and integral rheology model to simulate the filling process and more complex viscous phenomena.

**Author Contributions:** Conceptualization, A.R.-C. and M.C.; methodology, J.G., M.C. and E.A.; software, M.C and M.L.; validation, J.G., M.C. and M.L.; formal analysis, J.G., M.C. and M.L.; investigation, J.G., M.C. and M.L.; resources, A.R.-C. and E.A.; data curation, J.G., M.C. and M.L.; writing—original draft preparation, A.R.-C. and M.C.; writing—review and editing, A.R.-C. and M.L.; visualization, A.R.-C. and J.A.M.-P.; supervision, A.R.-C. and J.A.M.-P.; project administration, A.R.-C.; funding acquisition, A.R.-C. and J.A.M.-P. All authors have read and agreed to the published version of the manuscript.

**Funding:** This research received no external funding.

**Institutional Review Board Statement:** Not applicable.

**Informed Consent Statement:** Not applicable.

**Data Availability Statement:** Not applicable.

**Acknowledgments:** The authors would like to express their deepest gratitude to the Center of Nanotechnology Research and Development (CIDNA) and Laboratory of Testing Materials (LEMAT) for supporting our study with material characterization.

**Conflicts of Interest:** The authors declare no conflict of interest.

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
