# Peer review of "Evaluation of Processing Conditions in the Performance of Purging Compounds for Polypropylene Injection Molding"

_jmmp, doi:10.3390/jmmp7010031_

Round 1

Reviewer 1 Report

The article presents an interesting topic of polypropylene injection molding processing.

The article can be published after correction:

- The literature review needs to be supplemented with topics related to the researched materials.

- The literature is sufficient, but needs a slight addition.

- The article does not describe in detail all the parameters of the injection proces.

In particular, the injection process time, cooling time, mold temperature should be supplemented.

Changes in properties due to process parameters should be described in more detail.

- There is no detailed description of the manufacturing methodology of the tested materials: machines, devices, parameters. How were the mixtures made and how were the test samples made?

- The article presents very interesting research results, but the occurring phenomena are not fully explained on their basis. The description should be extended in this respect.

Reviewer 2 Report

Very up-to-date and, above all, useful research topics, unheard of in scientific publications. Thanks authors for that idea of research. Some suggestions and comments regarding the manuscript and conducted research:

Line 81 - Point the properties and description of tested materials in the table. 

Line 88 - It should by titled: Used research methods and equipment. Because authors not characterized the materials but describe mehods and used equipmentLine 150 - better will be Purging sample preparation

line 152 - lack of screw diameter (probably 75mm)

line 155 - Wine PP ... - better Red pigmented PP

line 157 - feeding throat - better feeding zone or hopper 

line 159 - change 15kg*cm2 in to bar or MPa (if you present injection pressure in bar you should present back pressure in the same unit) 

Figure 1 - why some of the mouldings are not fully injected 

Figure 1 - Was it results of experiment in one temperature? (220, 240 or 260)

line 183 - again 3.1 Characterization  - not necessary

In the results TGA analysis we can see that there are high content of fillers in PP-MP and PP-CP Filler materials. The microscopic observations of residuals of fillers obtained after the TGA analysis will be valuable for the investigations.  The phisical structure has a biggest influence on the rheological properties observed during the flow into the injection unit and mould cavity. 

Figure 4 - The PP virgin should be also present on thermograms

Figure 5 - Why Torque results for PE-CP in all temperatures are so unstable versus PP-MP

Systematize all results

Reviewer 3 Report

In this work, the authors compare purging compounds used in injection molding, primarily focusing on PP based mechanical purging mixture and a PE based chemical purging mixture. Experimental characterization and analysis includes Fourier Transform Infrared Spectroscopy and X-ray Diffraction, TGA analysis, DSC analysis, rheological analysis and a statistical understanding of specific energy consumption. The authors do a good job in their experimental presentation and reasoning. They present a nuanced analysis of the relative merits of both types of purging mixtures. Some minor questions remain regarding the statistical analysis. While the reviewer finds the quality of the paper acceptable, they recommend that the authors work to improve the readability of the paper with minor changes to the introduction and content structure.

1.       Line 22: A non-abbreviated form of PP presented in the first time it is mentioned would be recommended.

2.       Line 23: On the hand?

3.       The authors present an introduction on current industrial practices regarding purging compounds used during injection molding. While this seems relevant, the final paragraph of the introduction seems abrupt. The motivation for polyolefin based CPCs has not been developed and seems disconnected from the rest of the introduction. It is suggested that the authors make an effort to connect the earlier sections of the introduction (mechanical vs chemical purging, deviation from methodology, SEC etc.) to the proposed CPCs studied in this work. Questions that a reader may have at this juncture, and which preferably should have been addressed in the introduction are – why is this study focusing on polyolefin based CPCs? Are other form of purging mixtures already well studied? What is the relative performance of CPCs compared to other methods – which may motivate the current study?

4.       Line 162: How many data points were available for an ANOVA to be conducted? How is the purging time calculated? Are the temperature profiles in table 2 optimized for minimum downtime for each compound used? The reviewer suggests adding the results of the ANOVA as an appendix.

5.       Line 333: Was the p-value 0? The authors could specify beyond 3 decimals using scientific notation.

6.       Line 348: “Two purging … wax and surfactants.” The reviewer recommends having an abridged version of this section within the introduction. Perhaps somewhere in the last paragraph of the introduction section. It makes the readability of the paper a whole lot better after understanding the proposed comparisons.
